# Physics-Informed Coarsening for Multigrid Graph Neural Surrogates

**Amir Bazzi** [1][2]  **Ramy Nemer** [1]  **José Alves** [3]  **Elie Hachem** [1]

## Abstract

Learning-based surrogates for partial differential equations have recently matched the accuracy of classical solvers while achieving orders-of-magnitude speedups, predominantly in fluid settings and structured geometries. In contrast, robust surrogates for deformable solids remain underexplored, despite the presence of nonlinear elasticity, plasticity, and transient behavior that challenge standard architectures. We introduce a multigrid graph neural network for solid mechanics that couples an *encoder-processor-decoder* backbone with a physics-informed coarsening strategy. Instead of downsampling via geometric heuristics, our method scores nodes using a residual-based measure of local physical activity and preferentially retains regions of high strain or stress concentration, allocating multiscale capacity where it is most needed. This preserves long-range interactions through hierarchical message passing while improving stability over long rollouts. We evaluate on multiple datasets covering linear, nonlinear, and transient regimes, and observe consistent gains in accuracy and rollout stability compared to standard sampling baselines. Our results highlight the importance of physics-informed coarsening for scalable surrogate modeling in solid mechanics. Project page: https://sites.google.com/view/physics-informed-coarsening.

## 1. Introduction

Partial differential equations (PDEs) are central to modeling a wide range of natural and engineering systems, including turbulence and solid deformation (Launder & Spalding, 1983; Zhu & Zacharia, 1996). In industrial manufactur-ing, accurately solving these PDEs is critical for design and process optimization, but typically requires computationally expensive numerical methods such as the finite element method FEM (Belytschko et al., 1984; Zienkiewicz & Taylor, 2005). These simulations often involve complex geometries and require fine mesh discretizations to capture localized physical effects, which significantly increases computational cost (Solanki et al., 2003). As a consequence, high-fidelity industrial simulations can take between several days to weeks, limiting their usability in time-critical engineering workflows. This challenge is further amplified in solid mechanics, where strong nonlinearities (e.g., large deformations, plasticity, or contact) make the dynamics particularly difficult to simulate efficiently and robustly (Badia et al., 2021; Bird et al., 2024).

Despite the success of numerical solvers, their cost has motivated learning based surrogate models for accelerating PDE simulations (Chen et al., 2019; Mo et al., 2019). However, most existing benchmarks and models emphasize canonical fluid-dynamics or idealized settings (Raissi et al., 2019; Chen et al., 2021; Li et al., 2023). In contrast, practical solid mechanics simulations involve fully 3D unstructured meshes and strong nonlinearities (e.g., large deformations, plasticity, contact), making accurate long-horizon prediction substantially more challenging. (Libao et al., 2023; Gladstone et al., 2023; Lin et al., 2024)

We propose a multigrid graph neural network (GNN) architecture for mesh based solid mechanics simulation. Multigrid structures have demonstrated strong efficiency in recent learning based simulators (Garnier et al., 2024; Cao et al., 2022; Taghibakhshi et al., 2023) and are inspired by classical numerical solvers, finite element method solvers more specifically, where coarse-to-fine representations accelerate convergence and propagate global information. A key design choice in such hierarchical models is the downsampling/upsampling strategy, i.e., which nodes are retained at coarse levels and how coarse features are interpolated back to the fine graph. Unlike prior approaches relying on geometric heuristics or purely learned importance scores, our method introduces a *physics-informed sampling* mechanism that selects nodes based on a physical residual score derived from the governing solid mechanics equations. This biases the multigrid hierarchy toward dynamically critical regions (e.g., large deformation or stress-concentration zones),

---

[*]Equal contribution  [1]CEMEF, Mines Paris – PSL, Sophia Antipolis, France  [2]Aubert et Duval, France  [3]Transvalor, Mougins, France. Correspondence to: Amir Bazzi <amir.bazzi@minesparis.psl.eu>.

*Proceedings of the 43rd International Conference on Machine Learning*, Seoul, South Korea. PMLR 306, 2026. Copyright 2026 by the author(s).

thereby improving accuracy and long-term stability under complex solid regimes. In addition, we introduce two new benchmark datasets that target realistic solid mechanics regimes and cover a wide range of behaviors, including non-linear elasticity, plasticity, and transient dynamics. Figure 1 illustrates representative trajectories from these datasets, highlighting the diversity of deformations and physical conditions considered in this work.

The main contributions of this paper are:

- We propose a multigrid graph neural network surrogate model tailored to mesh based solid mechanics simulations, leveraging coarse-to-fine message passing on unstructured 3D meshes.

- We introduce a physics-informed sampling strategy for multigrid coarsening, selecting downsampled nodes based on a physical residuals, thereby focusing model capacity on dynamically critical regions.

- We release two novel benchmark datasets for learned solid mechanics, spanning diverse regimes including nonlinear elasticity, elasto-plasticity, and transient dynamics, complementing existing benchmarks.

- We demonstrate the effectiveness of the proposed method through extensive experiments across multiple datasets and regimes, showing improved long-horizon rollout accuracy compared to baseline sampling and surrogate approaches.

## 2. Related work

A significant amount of work has recently emerged to replace or accelerate classical numerical solvers by learning surrogate models for partial differential equations (PDEs) (Khoo et al., 2021). In particular, Graph Neural Networks (GNNs) and Transformers (Scarselli et al., 2008; Vaswani et al., 2017) have been widely explored with various architectures, and several models achieved impressive speedups on complex geometries. In parallel, operator learning has been extensively studied to solve PDEs by directly learning the mapping between input functions and their corresponding solutions (Li et al., 2021; Wen et al., 2022; Cao et al., 2024; Li et al., 2023; Wang & Wang, 2024).

A representative example of GNN based simulation is the Encode-Process-Decode framework applied to mesh based physical dynamics, such as MeshGraphNet (Sanchez-Gonzalez et al., 2020; Pfaff et al., 2021), which demonstrated strong performance across a range of benchmark geometries. This architecture has inspired several follow-up works extending message passing for improved stability, generalization, and scalability. (Lucchetti et al., 2025)

Transformers have also been adapted to physical simulation due to their ability to capture long range dependencies and global interactions (Han et al., 2022). Despite promising results, Transformer based surrogates may incur some limitations, such as lack of positional awareness and computational complexity of $\mathcal{O}(n^2)$ (Garnier et al., 2025), and often rely on downsampling or hierarchical mechanisms to remain tractable (Yu et al., 2023). UNISOMA (Tao et al., 2025) reduces the cost of attention by compressing each object into a fixed number of slice tokens, but this compression may discard fine local details near sharp contact or stress-concentration regions. This motivates scalable multilevel approaches that preserve local mechanics while enabling long-range propagation.

On the other hand, multiscale processing is a classical idea in numerical PDE solvers (Bank et al., 1988; Brandt, 1977), where coarse and fine representations are coupled to accelerate convergence and propagate global information efficiently.

Inspired by this principle, recent GNN works introduced multigrid-like architectures, drawing an analogy with U-Net (Ronneberger et al., 2015), to propagate information over long spatial ranges while avoiding issues such as over-smoothing caused by repeated local aggregation in deep message-passing networks.

For instance, MultiScale MeshGraphNets (Fortunato et al., 2022) leverages both fine and coarse mesh resolutions; however, constructing and maintaining multiple meshes introduces additional complexity.

Graph coarsening and downsampling are crucial in hierarchical simulators, as the selected coarse nodes must retain both the global structure and the physically active regions of the domain. Common approaches include random sampling and geometric heuristics such as farthest point sampling (FPS) (Qi et al., 2017).

In particular, (Cao et al., 2022) introduces a topology-driven bi-stride pooling strategy to construct multiscale graphs while preserving connectivity, whereas other multigrid-inspired GNNs (Lino et al., 2021; Garnier et al., 2024) explicitly couple coarse and fine graph representations through downsampling and upsampling operations, using learned importance scores to guide node selection across scales.

## 3. Background

### 3.1. Governing Equations and solid dynamics

**Partial differential equation.** A partial differential equation is an equation that involves an unknown function of $n \geq 2$ variables and (some of) its partial derivatives. These PDEs can be described by one or a set of equations that describe the evolution of a field in space and time:

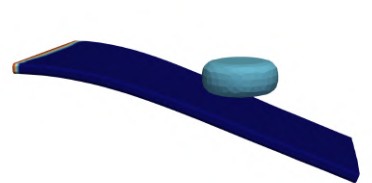

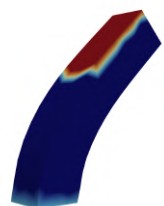

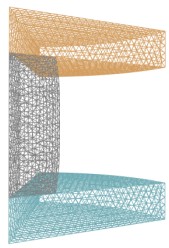

*(a)* DEFORMINGPLATE (quasi-static hyper-elastic).
*(b)* BEAMSIMPLE (nonlinear hyper-elasticity).
*(c)* SPINDLEUPSETTING (plasticity + contact).

*Figure 1.* Representative trajectories from our three solid mechanics datasets, spanning quasi-static hyperelastic deformation, nonlinear elastic dynamics, and elasto-plastic forming with contact. All datasets are defined on 3D unstructured meshes.[1]

$$\mathcal{L}(\boldsymbol{u}, \nabla^1 \boldsymbol{u}, \ldots, \nabla^\ell \boldsymbol{u}, \partial_t^1 \boldsymbol{u}, \ldots, \partial_t^k \boldsymbol{u}, \boldsymbol{x}, t) = 0.$$

Here, $\mathcal{L}$ denotes an operator-valued mapping taking values in $\mathbb{R}^d$. We use $\nabla^\ell \boldsymbol{u}$ to denote spatial derivatives up to order $\ell$ and $\partial_t^k \boldsymbol{u}$ temporal derivatives up to order $k$.

Let $T > 0$ denote the final time horizon and $\Omega \subset \mathbb{R}^3$ the spatial domain. A function $\boldsymbol{u} : [0, T] \times \Omega \to \mathbb{R}^d$ belonging to $\mathcal{C}^\ell([0, T] \times \Omega)$ and satisfying the PDE pointwise is said to be a *classical solution*.

The formulation is completed by specifying appropriate initial conditions at $t = 0$, together with boundary conditions expressed through an operator $\mathcal{B}[\boldsymbol{u}]$ imposed on the boundary $\partial \Omega$.

This equation can vary from one case to another and may even be unknown in some cases.

We primarily work with the continuum solid mechanics equation, governed by the balance of linear momentum, which is considered the fundamental governing equation for all solid mechanics:

$$\rho \ddot{\boldsymbol{u}} = \nabla \cdot \boldsymbol{\sigma} + \rho \mathbf{b}, \tag{1}$$

where $\boldsymbol{u}(\boldsymbol{x}, t)$ is the displacement field, $\rho$ is the density, $\boldsymbol{\sigma}$ is the Cauchy stress tensor, and $\mathbf{b}$ denotes the body force per unit mass. The constitutive model defining $\boldsymbol{\sigma}$ depends on the physical regime (hyperelasticity, nonlinear elasticity, plasticity), as detailed in section B.

**Solid mechanics and physics residuals.** We consider continuum solid mechanics, where the deformation and motion of solids are governed by the balance of linear momentum together with constitutive relations describing material behavior. Depending on the regime, materials may exhibit elastic (recoverable) or plastic (permanent) deformation, leading to nonlinear governing equations and distinct physical responses. To guide physics-informed modeling across these regimes, we define a nodal physics score using physical fields. In particular, we evaluate a nodal mechanical residual from the model-predicted fields by combining stress divergence, inertia when relevant, and body-force terms. The stress-divergence term is obtained using a fixed mesh based derivative reconstruction, and the residual is recomputed autoregressively at each timestep. This allows the model to leverage physically consistent information while remaining applicable across multiple solid mechanics behaviors. Full implementation details are provided in section B.

## 4. Method

### 4.1. Problem formulation

We consider learning surrogate time integration for solid mechanics simulations defined on deformable meshes. At each time step $t$, the mesh is represented as a graph $\mathcal{G} = (\mathcal{V}, \mathcal{E})$, where nodes correspond to mesh vertices and edges represent mesh connectivity. The physical state is given by a nodal field

$$\boldsymbol{u}^t : \mathcal{V} \to \mathbb{R}^d,$$

where $\boldsymbol{u}_i^t$ denotes the state at node $i$ (e.g., displacement, velocity, or position).

Our goal is to learn a surrogate operator that advances the system by one time step. Following common practice in learned simulators, we predict an incremental update

$$\boldsymbol{u}_{t+1} = \boldsymbol{u}_t + \Phi_\theta(\boldsymbol{u}_t, \mathcal{G}), \tag{2}$$

where $\Phi_\theta$ is implemented as a graph neural network. This residual formulation mirrors classical time integration schemes and improves stability for long rollouts.

The main challenge in solid mechanics is that dynamics often involve large deformations, strong nonlinearities (e.g., plasticity and contact), and unstructured 3D meshes. Therefore, $\Phi_\theta$ must capture both local interactions and long-range global coupling efficiently. To this end, we propose a multi-grid GNN architecture driven by physics-informed node selection.

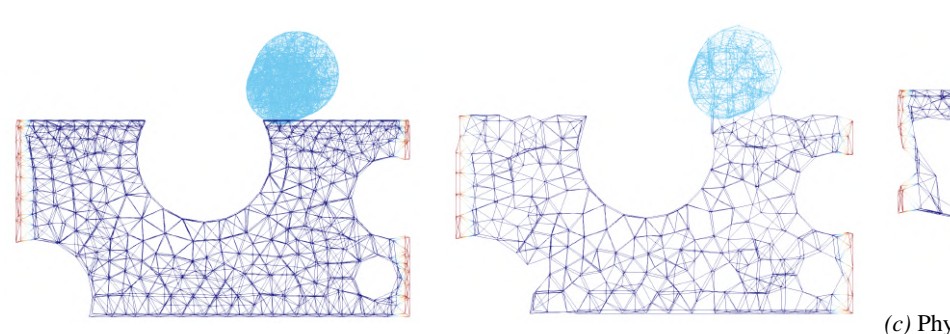

*(a)* No downsampling (reference mesh).  *(b)* FPS sampling (w/ remeshing).  *(c)* Physics-informed sampling (w/ remeshing).

*Figure 2.* Comparison of coarse node selection and resulting mesh structures on *DeformingPlate* at timestep 20, with node types shown in color. This is an *all nodes* sampling strategy.

## 4.2. Architecture overview

Our model follows an encoder-processor-decoder design commonly used for mesh based simulation (Pfaff et al., 2021). The encoder $\phi_{\text{enc}}$ lifts node features into a latent space, the processor performs message passing and multiresolution refinement, and the decoder $\phi_{\text{dec}}$ maps latent features back to the physical state.

Unlike single-scale GNN simulators, our processor explicitly builds a coarse representation using physics-informed sampling. This enables information propagation across long spatial ranges while preserving computational tractability. Figure 3

## 4.3. Encoder and decoder

The encoder applies a pointwise MLP to lift raw node features into a latent dimension $h$:

$$\phi_{\text{enc}} : \mathbb{R}^d \to \mathbb{R}^h. \tag{3}$$

The decoder maps final latent features back to the physical space using another pointwise MLP:

$$\phi_{\text{dec}} : \mathbb{R}^h \to \mathbb{R}^3. \tag{4}$$

## 4.4. Processor blocks

The processor consists of message passing blocks combined with downsampling/upsampling stages that define the multigrid hierarchy. At a high level, it alternates between: (i) fine-scale message passing, (ii) coarse processing after downsampling, and (iii) refinement after upsampling.

**GraphNet block.** We use the GraphNet update rule introduced in MeshGraphNets (Pfaff et al., 2021). Given node features $\mathbf{v}_i \in \mathbb{R}^h$ and edge features $\mathbf{e}_k \in \mathbb{R}^h$, edges are first updated, then aggregated to update nodes: For each directed edge $k = (s_k, r_k)$, where $s_k$ is the sender node and $r_k$ is the receiver node, the edge feature is updated using the current edge feature and the features of its incident nodes:

$$\mathbf{e}_k' = f_e(\mathbf{e}_k, \mathbf{v}_{s_k}, \mathbf{v}_{r_k}). \tag{5}$$

The updated edge features are then aggregated at each receiver node $r$:

$$\bar{\mathbf{e}}_r' = \sum_{k : r_k = r} \mathbf{e}_k'. \tag{6}$$

Finally, the node feature is updated using its previous feature and the aggregated incoming message:

$$\mathbf{v}_r' = f_v(\mathbf{v}_r, \bar{\mathbf{e}}_r'). \tag{7}$$

Here, $f_e$ and $f_v$ are learnable MLPs shared across all edges and nodes, respectively, and implemented with layer normalization and residual connections following (Pfaff et al., 2021). This defines an operator

$$\text{GN} : \mathcal{G}^{n \times h} \to \tilde{\mathcal{G}}^{n \times h}.$$

**Downsampling block.** Given a processed fine latent graph $\tilde{\mathcal{G}}^{n \times h}$, the downsampling block constructs a coarse graph by selecting the most informative nodes for global propagation. To guide this selection, we first decode latent node features into physical quantities using the decoder $\mathcal{D}$:

$$\mathcal{D} : \mathbb{R}^h \to \mathbb{R}^3.$$

From these decoded quantities, we compute a physics residual score $s_i \in \mathbb{R}$ for each node $i$, following solid mechanics residuals, measuring local physical activity.

Coarse nodes are then selected using either a deterministic ranking or probabilistic sampling operator:

$$\mathcal{S}(\mathbf{s}) : \mathbb{R}^n \to \{1, \ldots, n\}^{n_s}, \qquad n_s < n.$$

The resulting coarse latent graph inherits node embeddings from the fine graph:

$$\text{DN} : \tilde{\mathcal{G}}^{n \times h} \to \mathcal{G}_c^{n_s \times h}.$$

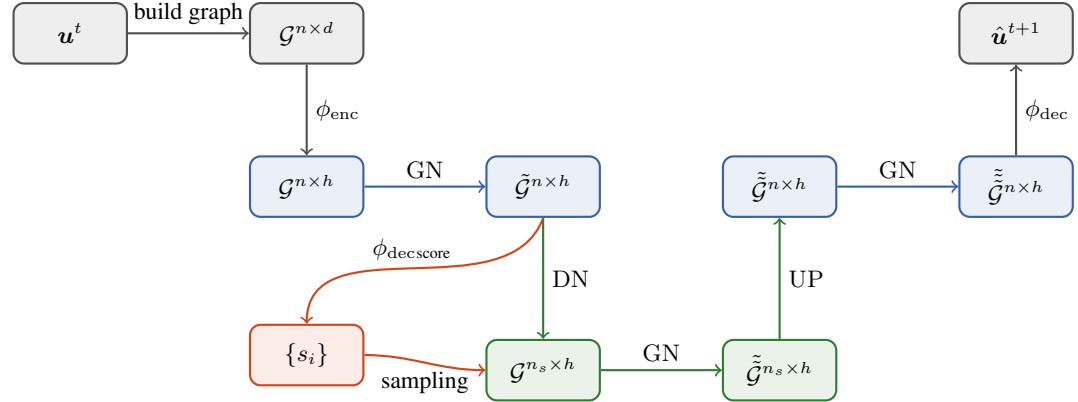

*Figure 3.* Compact multigrid encoder-processor-decoder architecture with physics-informed sampling. Color encodes semantic roles: physical space (gray), latent message passing (blue), multigrid hierarchy (green), and physics based scoring and sampling (orange).

**Upsampling block.** The upsampling block transfers information back from the processed coarse graph to the fine resolution. Given a coarse representation $\tilde{\mathcal{G}}_c^{n_s \times 2h}$, we interpolate coarse embeddings onto the fine nodes using $k$-nearest neighbors (KNN) in physical space:

$$\text{UP}: \tilde{\mathcal{G}}_c^{n_s \times h} \rightarrow \tilde{\mathcal{G}}^{n \times h}.$$

The interpolated features are fused with fine-level embeddings and refined through additional GraphNet blocks, enabling both global context (via coarse processing) and local accuracy (via fine refinement).

**Processor operators.** In summary, our processor alternates three operators:

$$
\begin{aligned}
\text{GN} &: \mathcal{G}^{n \times h} \rightarrow \tilde{\mathcal{G}}^{n \times h}, \\
\text{DN} &: \tilde{\mathcal{G}}^{n \times h} \rightarrow \mathcal{G}_c^{n_s \times h}, \\
\text{UP} &: \tilde{\mathcal{G}}_c^{n_s \times h} \rightarrow \tilde{\mathcal{G}}^{n \times h}.
\end{aligned}
\tag{8}
$$

When composed, they define a multiscale message-passing schedule:

$$\mathcal{G} \rightarrow \tilde{\mathcal{G}} \rightarrow \mathcal{G}_c \rightarrow \tilde{\mathcal{G}},$$

where the coarse stage increases the effective receptive field and improves long-horizon stability.

### 4.5. Physics-informed coarsening

A central component of hierarchical (multigrid) graph simulators is the coarsening operator, which defines how the fine mesh graph is reduced into a coarse representation. Most existing multigrid simulators rely on geometric heuristics such as farthest point sampling (FPS) or learned attention scores. However, our method relies on choosing the most informative regions, i.e., regions exhibiting strong physical activity such as stress concentration zones, large deformation regions, boundary transitions, or contact interfaces. Motivated

by this, we propose a physics-informed coarsening strategy that explicitly selects coarse nodes based on the local violation of the governing equations.

**Residual based physics score.** Let $\tilde{\mathcal{G}}^{n \times h}$ denote a processed latent graph at the fine resolution. To evaluate node importance, we first decode latent node embeddings to the physical space using the decoder $\phi_{\text{dec}}$:

$$\hat{\boldsymbol{u}}^t = \phi_{\text{dec}}(\tilde{\mathcal{G}}). \tag{9}$$

We then compute a nodal physics residual $\mathbf{r}_i^t$ for each node $i$ from the predicted state. For transient solid dynamics, this residual is written as a discrete mechanical imbalance:

$$\mathbf{r}_i^t = \rho_i \ddot{\hat{\boldsymbol{u}}}_i^t - \left(\nabla_h \cdot \hat{\boldsymbol{\sigma}}^t\right)_i - \rho_i \mathbf{b}_i^t, \tag{10}$$

where $\nabla_h \cdot$ denotes a fixed mesh based discrete divergence operator. In quasi-static regimes, the inertial term is omitted, yielding the equilibrium residual

$$\mathbf{r}_i^t = -\left(\nabla_h \cdot \hat{\boldsymbol{\sigma}}^t\right)_i - \rho_i \mathbf{b}_i^t. \tag{11}$$

The scalar node score is defined as

$$s_i^t = \|\mathbf{r}_i^t\|_2. \tag{12}$$

Large values of $s_i^t$ indicate nodes that contribute strongly to the predicted violation of local mechanical balance. Thus, the score acts as an a posteriori indicator of mechanically active or difficult regions, such as stress concentrations, boundary transitions, large deformation zones, or contact-induced imbalance. Full residual expressions for each dataset are provided in Appendix B.

Intuitively, the residual magnitude acts as an indicator of local mechanical difficulty. Regions with large residuals correspond to areas where the predicted state strongly violates the governing balance laws, typically near stress concentrations,

*Table 1.* Summary of datasets used in our experiments.

| DATASET | SOLVER | # NODES | DIM. | # TRAJ | # STEPS | $\Delta t_s$ |
|---------|--------|---------|------|--------|---------|-----------|
| DEFORMINGPLATE | COMSOL | 1K | 3D | 100 | 400 | – |
| BEAMSIMPLE | CIMLIB | 500 | 3D | 150 | 500 | 0.01 |
| BENDINGBEAM | CIMLIB | 1.2K | 3D | 140 | 500 | 0.01 |
| SPINDLEUPSETTING | FORGE | 9K | 3D | 100 | 83 | 0.0009 |

contact interfaces, boundary transitions, or large deformation zones. Allocating coarse level capacity to these regions improves the propagation of physically important information across scales and supports more accurate long-range corrections during message passing. This idea is closely related to adaptive refinement strategies in classical finite element methods, where residual-based error indicators are commonly used to identify regions requiring increased resolution or computational focus.

**Physics-informed node selection.** The node scores are collected into a score vector $s \in \mathbb{R}^n$. Given a target coarse size $n_s < n$, we select a subset of coarse nodes $\mathcal{V}_c \subset \mathcal{V}$ using either a deterministic or stochastic strategy.

**TopK selection** deterministically keeps the nodes with highest residual scores:

$$\mathcal{V}_c = \arg \max_{\substack{\mathcal{S} \subset \mathcal{V} \\ |\mathcal{S}|=n_s}} \sum_{i \in \mathcal{S}} s_i^t, \qquad (13)$$

which is implemented by

$$\mathcal{I} = \text{TopK}(s^t, n_s), \qquad \mathcal{V}_c = \{v_i \mid i \in \mathcal{I}\}. \qquad (14)$$

**Probabilistic selection** samples indices from a categorical distribution biased by normalized scores:

$$p_i = \frac{s_i}{\sum_{j=1}^n s_j}, \qquad i = 1, \ldots, n, \qquad (15)$$

which introduces controlled stochasticity while still focusing the hierarchy on physically active regions.

In both cases, selected nodes define the coarse graph, and coarse node embeddings are inherited from their fine counterparts.

**Remeshing vs inherited connectivity.** Once the coarse node set $\mathcal{V}_c$ is selected, we must define the coarse edge set $\mathcal{E}$. We consider two strategies. In the **inherited connectivity** setting, the coarse graph is obtained as the induced subgraph of the fine mesh:

$$\mathcal{E}_c = \{(i,j) \in \mathcal{E} \mid i \in \mathcal{V}_c, \ j \in \mathcal{V}_c\}. \qquad (16)$$

In the **remeshing** setting, coarse connectivity is rebuilt directly on the sampled nodes using a $k$-nearest-neighbors

graph in Euclidean space (Cover & Hart, 1967). Remeshing typically yields a more consistent coarse topology and improves long-range propagation, but introduces additional preprocessing and may not preserve the exact original fine-mesh adjacency Figure 2c.

**Effect of physics-informed coarsening.** Physics-informed coarsening should not be viewed purely as a computational optimization. While downsampling reduces the number of nodes at the coarse level, the multigrid design shifts representational capacity toward selected regions via additional processing stages. Using physics based scores biases the coarse graph toward dynamically active regions, which improves long-range information propagation and supports accurate reconstruction of localized deformations after upsampling. A qualitative comparison with FPS is provided in Figure 2.

## 5. Datasets

We evaluate our method on three solid mechanics datasets designed to cover a broad spectrum of physical regimes, ranging from quasi-static nonlinear elasticity to transient dynamics and industrial-scale plastic deformation. The goal is twofold: (i) to validate that the proposed multigrid surrogate can robustly operate on fully 3D unstructured meshes, and (ii) to demonstrate its ability to generalize across diverse solid dynamics beyond canonical fluid benchmarks.

Table 1 summarizes the main characteristics of each dataset, including solver origin, mesh resolution, and temporal discretization. All datasets are represented as mesh based graphs with nodal physical states, enabling evaluation under a unified surrogate learning framework.

**DEFORMINGPLATE (quasi-static hyperelasticity).** This benchmark, introduced by MeshGraphNets (Pfaff et al., 2021), models nonlinear deformation of a thin structure governed by a hyperelastic constitutive law ( Figure 1a). The system evolves in a *quasi-static* regime, where inertial terms are neglected and equilibrium is solved at each time step. Despite the absence of transient effects, the dataset contains large deformations and nonlinear stress-strain behavior, making long-horizon stability challenging.

**BEAMSIMPLE / BENDINGBEAM (nonlinear elasticity).**
These datasets represent deformable beams undergoing *nonlinear elastic* deformation ( Figure 1b). They are simulated using CimLib (Nemer et al., 2021) and exhibit smooth but highly nonlinear geometric behavior in 3D, with deformation propagating over long spatial scales. These datasets particularly stress the ability of surrogate models to capture global coupling and geometric nonlinearity on unstructured meshes.

**SPINDLEUPSETTING (industrial elasto-plasticity).**
SPINDLEUPSETTING is an industrial-scale metal forming process simulated using the FORGE® finite element software ( Figure 1c). It features large plastic deformation, strong nonlinearities, and complex boundary interactions typical of industrial forming applications. The process involves tool–workpiece contact, which we represent in the graph using explicit *world edges* encoding interactions with external bodies, as detailed in Appendix subsection A.3.

## 6. Main Results

### 6.1. Experimental setup

We first evaluate our model on the *DeformingPlate* dataset, for an ablation study and analyzing the performance of the models.

**Predictive task.** All methods are trained as surrogate time integrators: given a mesh state at time step $t$, the model predicts an incremental update to advance the system to $t + 1$. We assess performance with two primary metrics: (i) 1-step RMSE, computed between the ground-truth next state and the model prediction at the next step, and (ii) rollout RMSE, computed over long-horizon rollouts by recursively feeding model predictions back as inputs.

**Training.** All models are trained for 30 epochs, corresponding to approximately $10^6$ optimization steps depending on the dataset size and batch size. We use the AdamW optimizer (Loshchilov & Hutter, 2019) with the same global training protocol for all baselines, and tune model-specific hyperparameters on a validation set. Experiments are conducted on NVIDIA A100 GPUs. To reduce variance and ensure fair comparison, we run each configuration with multiple random seeds (between 3 and 5) and report the mean performance.

**Downsampling ratio.** For all multigrid based models, we downsample to a coarse graph containing 50% of the fine nodes at each downsampling stage. This ratio is fixed across ablations.

### 6.2. Ablation Study: DeformingPlate comparison

We compare our physics-informed multigrid GNN against strong learned simulators on *DeformingPlate*, including MeshGraphNets, BSMS, Transolver++, Multi-Scale GNN, HCMT, UNISOMA, and an Encode-Transformer-Decode hybrid baseline. All methods are trained under the same protocol (identical training split, number of epochs, and evaluation horizon) details in subsection A.1, and each model is tuned on a validation set to ensure a fair comparison. Table 2 reports rollout and one-step prediction errors, together with training cost, highlighting both accuracy and efficiency trade-offs across methods.

### 6.3. Multigrid + sampling study: physics-informed node selection

A central design choice in hierarchical graph simulators is the *coarsening strategy*, i.e., how to select the subset of nodes that define the coarse graph during downsampling. Because the coarse level is responsible for propagating global information efficiently, the quality of selected nodes directly impacts both prediction accuracy and long horizon rollout stability. In this section, we isolate the effect of node selection by comparing different sampling strategies within the same multigrid architecture and training protocol.

**Results.** Table 3 reports rollout and one-step errors for all sampling schemes. Across all metrics, our physics-informed sampling consistently outperforms geometric (FPS) and learned (attention based) alternatives. In particular, we observe the largest gains in long-horizon rollouts, indicating that allocating coarse-level capacity toward physically active regions (e.g., stress concentrations, high deformation zones, boundary transitions) is critical for stable autoregressive prediction. These results support our central hypothesis: *incorporating physical residual information into multigrid coarsening provides a more effective criterion than geometry alone for identifying mechanically important regions in solid mechanics simulations.*

**Sampling strategies.** We compare the following approaches: (i) **Random sampling**, which selects coarse nodes uniformly at random; (ii) **Farthest Point Sampling (FPS)** (Qi et al., 2017), which selects nodes to maximize geometric coverage; (iii) **attention based sampling** (Garnier et al., 2024), where nodes are selected based on learned importance scores; and (iv) our proposed **physics-informed sampling**, where nodes are selected based on a physical activity score derived from governing-equation residuals (see section B). All methods use the same downsampling ratio (50% coarse nodes) to control for coarse-graph capacity and isolate the effect of the sampling strategy.

*Table 2.* Main benchmark comparison on *DeformingPlate*. Lower is better.

| Method | Rollout RMSE ($\times 10^{-3}$)↓ | 1-Step RMSE ($\times 10^{-3}$)↓ | #Parameters↓ |
|---|---|---|---|
| MeshGraphNets (Pfaff et al., 2021) | 12.75 | 0.10 | 2.8M |
| BSMS (Cao et al., 2022) | 16.60 | 0.15 | 2.1M |
| Transolver++ (Luo et al., 2025) | 29.80 | 1.00 | 722K |
| Transformer GNN | 24.97 | 1.20 | 3.5M |
| Multi-Scale GNN (Lino et al., 2021) | 15.7 | 0.10 | 3.1M |
| HCMT (Yu et al., 2023) | 12.97 | 0.14 | 2.53M |
| UNISOMA (Tao et al., 2025) | 11.46 | 0.16 | 2.85M |
| **Ours (Physics-informed Multigrid GNN)** | **6.50** | **0.095** | 2.9M |

*Table 3.* Comparison of multigrid sampling strategies on *DeformingPlate*.

| Sampling strategy | Rollout RMSE ($\times 10^{-3}$)↓ | 1-Step RMSE ($\times 10^{-5}$)↓ |
|---|---|---|
| BSMS (Cao et al., 2022) | 16.60 | 15 |
| FPS (w/o remeshing) (Qi et al., 2017) | 15.0 | 10.31 |
| Attention (Garnier et al., 2024) | 8.1 | 17.10 |
| FPS (w/ remeshing) (Qi et al., 2017) | 8.0 | 9.74 |
| Physics-informed sampling w/ probabilistic sampling | 13.1 | 11.32 |
| **Physics-informed sampling w/ TopK (Ours)** | **6.5** | **9.57** |

**Decoder reuse for physics scoring.** One could wonder whether reusing the same decoder for state prediction and physics based scoring introduces bias or instability. Using a separate decoder for residual computation led to worse rollout accuracy and stability, indicating that decoder reuse encourages aligned and physically consistent representations.

**Coarse processor width (128 vs 256).** We study the impact of increasing the capacity of the coarse-level processor by using a wider MLP hidden dimension during the coarse message passing stage, inspired by the U-net structure (Ronneberger et al., 2015) As shown in Table 5, increasing the coarse MLP width does not significantly improve performance, suggesting that the coarse processor is already expressive enough to model our features.

**Normal nodes vs. all nodes scoring.** We further investigate whether physics-informed scoring should be computed using only *normal nodes* i.e., mesh nodes that are neither subject to Dirichlet boundary conditions nor involved in contact or kinematic constraints—or using *all nodes*, including boundary and constrained nodes. Figure 4 shows that scoring all nodes consistently yields lower error, indicating that constrained and boundary nodes also carry physically informative signals (e.g., reaction forces and stress concentrations) that are beneficial for constructing an effective multigrid hierarchy.

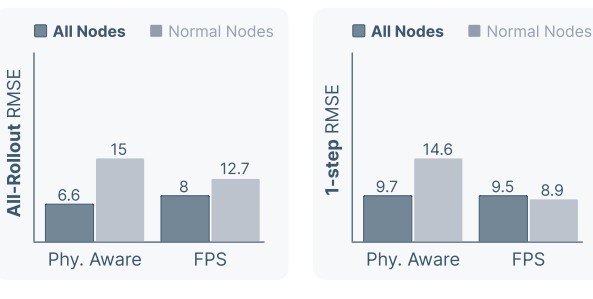

*(a)* Rollout RMSE.   *(b)* 1-step RMSE.

*Figure 4.* Ablation on physics-informed scoring: using all nodes vs only normal nodes for computing physics based scores, compared across sampling strategies (Physics-informed vs FPS). Lower is better.

**Results on other datasets.** To assess generalization beyond quasi-static deformation, we evaluate our approach on *BeamSimple* and *SpindleUpsetting*. Our multigrid model consistently improves rollout stability over MGN, highlighting the benefit of hierarchical processing in dynamic solid regimes. Residual based sampling performs comparably to, and sometimes better than, FPS, indicating that physics-informed node selection remains effective in transient nonlinear elasticity where deformation is localized and evolving. Results are shown in Table 4.

*Table 4.* Generalization performance on unseen datasets. Errors are averaged over the rollout horizon. Lower is better.

| Method | BeamSimple | | SpindleUpsetting | |
|---|---|---|---|---|
| | Rollout $(\times 10^{-1})\downarrow$ | 1-step $(\times 10^{-3})\downarrow$ | Rollout $(\times 10^{-1})\downarrow$ | 1-step $(\times 10^{-3})\downarrow$ |
| MeshGraphNet (Pfaff et al., 2021) | $1.72 \pm 0.28$ | $\mathbf{0.42 \pm 0.02}$ | $3.07 \pm 0.46$ | $11.92 \pm 0.47$ |
| FPS | $1.56 \pm 0.53$ | $0.48 \pm 0.02$ | $\mathbf{2.60 \pm 0.09}$ | $11.66 \pm 0.42$ |
| **Ours (Physics-informed Multigrid)** | $\mathbf{1.44 \pm 0.25}$ | $0.45 \pm 0.01$ | $2.96 \pm 0.26$ | $\mathbf{11.24 \pm 0.40}$ |

*Table 5.* Effect of coarse-level MLP width on *DeformingPlate*. Lower is better.

| Variant RMSE | Rollout $\downarrow$ | 1-Step $\downarrow$ |
|---|---|---|
| Coarse MLP $h_c = 128$ | $6.59 \times 10^{-3}$ | $9.5 \times 10^{-5}$ |
| Coarse MLP $h_c = 256$ | $15.72 \times 10^{-3}$ | $10.7 \times 10^{-5}$ |

## 7. Conclusion

In pursuit of practical learned simulators for solid mechanics, this paper presents a physics-informed multigrid graph neural network that improves long-horizon prediction on fully 3D unstructured meshes. Our key idea is to make multigrid coarsening *physics-informed*: instead of selecting coarse nodes using geometric coverage (e.g., FPS) or learned attention alone, we rank (or sample) nodes using a residual based score derived from the governing balance laws (and constitutive relations when applicable). This focuses coarse-level computation on dynamically critical regions such as stress concentrations, boundary transitions, and contact zones, while still enabling efficient global information propagation through coarse-to-fine message passing.

As a result, our model improves rollout stability and accuracy over strong baselines on DEFORMINGPLATE, DEFORMINGBEAM and SPINDLEUPSETTING and the same physics-informed coarsening principle transfers to nonlinear elastic beam dynamics. Beyond the method, we introduce two new benchmark datasets that broaden evaluation beyond quasi-static hyperelastic deformation, covering nonlinear elasticity and industrial elasto-plastic forming with contact. Together, these contributions move learned solid mechanics simulation closer to realistic engineering regimes where robustness, long-horizon stability, and scalability are essential.

### Limitations and perspective.

A limitation of our approach is that computing residual based scores can be nontrivial, especially for complex materials, distorted meshes, or simulations rich in contact. It requires derivative reconstruction and access to physically meaningful fields, which partially brings back ingredients from classical numerical solvers. Nevertheless, this also highlights the key idea of the paper: when such residuals can be approximated, they provide a principled criterion for identifying mechanically important nodes and guiding multigrid coarsening beyond purely geometric sampling.

## Impact statement

This paper presents work whose goal is to advance the field of machine learning, in particular learning based surrogate modeling for physical simulation. There are many potential societal consequences of this line of research, none of which we feel must be specifically highlighted here.

## Acknowledgements

I would like to express my sincere gratitude to Aubert et Duval for funding this project and for providing the industrial context, technical support, and resources that made this work possible. I am especially grateful to Paul Garnier for his continuous support, availability, and careful follow-up throughout this project.

I would also like to thank my supervisors, Elie Hachem, David Cardinaux, and Arjun Kalkur Matpadi Raghavendra, for their scientific guidance, insightful feedback, and constant support during this work.

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

# A. Additional Experimental Details

## A.1. Full ablation logs and seed variability

We report additional ablation and baseline-tuning experiments conducted on *DeformingPlate*. Each configuration was run with 1-5 random seeds. We report the mean and standard deviation across seeds for rollout RMSE and 1-step RMSE in our paper.

*Table 6.* Baseline tuning experiments on *DeformingPlate*. Reported values are mean ± std over 1-5 seeds.

| Model | MP layers | #heads | Hidden | Rollout RMSE ↓ | 1-step RMSE ↓ |
|---|---|---|---|---|---|
| MeshGraphNets (Pfaff et al., 2021) | 15 | – | 128 | $(1.28 \pm 0.41) \times 10^{-2}$ | $(9.99 \pm 0.06) \times 10^{-5}$ |
| MeshGraphNets (Pfaff et al., 2021) | 10 | – | 128 | $(1.30 \pm 0.33) \times 10^{-2}$ | $(1.07 \pm 0.09) \times 10^{-4}$ |
| MeshGraphNets (Pfaff et al., 2021) | 15 | – | 64 | $1.30 \times 10^{-2}$ | $(9.90 \pm 0.00) \times 10^{-5}$ |
| MeshGraphNets (Pfaff et al., 2021) | 10 | – | 64 | $(1.40 \pm 0.27) \times 10^{-2}$ | $(9.36 \pm 0.10) \times 10^{-5}$ |
| BSMS (Cao et al., 2022) | 3 | – | 128 | $1.66 \times 10^{-2}$ | $1.50 \times 10^{-4}$ |
| HCMT (Yu et al., 2023) | – | – | – | $1.297 \times 10^{-2}$ | $1.47 \times 10^{-4}$ |
| UNISOMA (Tao et al., 2025) | – | – | – | $1.146 \times 10^{-2}$ | $1.69 \times 10^{-4}$ |
| Transformer (Garnier et al., 2025) | 10 | 8 | 128 | $(2.86 \pm 0.36) \times 10^{-2}$ | $(1.21 \pm 0.01) \times 10^{-3}$ |
| Transolver++ (Luo et al., 2025) | 10 | 8 | 128 | $2.98 \times 10^{-2}$ | $1.00 \times 10^{-5}$ |

For the additional HCMT and UNISOMA baselines, we followed the configurations reported in the original papers whenever possible. For HCMT (Yu et al., 2023), we used the *DeformingPlate* setting with a total of $L = L_C + L_H = 15$ transformer blocks, split into $L_C = 10$ contact mesh transformer blocks and $L_H = 5$ hierarchical mesh transformer blocks, with hierarchy level $\lambda = 2$, hidden dimension 128, and 4 attention heads. For UNISOMA (Tao et al., 2025), we used the default configuration reported by the authors across experiments: 2 processor layers, hidden dimension 128, and 32 slice tokens.

*Table 7.* Additional ablations for the proposed multigrid model on *DeformingPlate*. Mean ± std over 1–5 seeds.

| Variant | Remeshing | Nodes | Rollout RMSE $(\times 10^{-2})$ ↓ | 1-step RMSE $(\times 10^{-4})$ ↓ |
|---|---|---|---|---|
| FPS | ✓ | All nodes | $0.80 \pm 0.31$ | $0.93 \pm 0.05$ |
| FPS | × | All nodes | 1.50 | 0.98 |
| FPS | × | Normal nodes | 1.84 | 1.03 |
| FPS (Sampling) | × | All nodes | 1.53 | 1.01 |
| Physics-informed (TopK) | ✓ | All nodes | $\mathbf{0.66 \pm 0.11}$ | $0.96 \pm 0.11$ |
| Physics-informed (TopK) | × | All nodes | 1.50 | 0.97 |
| Physics-informed (Sampling) | ✓ | All nodes | $1.31 \pm 0.38$ | $1.13 \pm 0.04$ |
| Physics-informed (Sampling) | ✓ | Normal nodes | 1.84 | **0.90** |
| Physics-informed (TopK) + MLP 256 | ✓ | All nodes | 1.57 | 1.07 |
| Physics-informed (TopK) + new decoder | ✓ | All nodes | 1.31 | 1.00 |

## A.2. Effect of noise injection on DeformingBeam

We study the impact of additive Gaussian noise injected during training on the *DeformingBeam* dataset. Noise injection is commonly used to improve robustness and stabilize long-horizon rollouts, but it may also degrade short-term prediction accuracy.

Table 8 reports the final validation rollout RMSE for different noise amplitudes. We observe that a moderate noise level of $2 \times 10^{-3}$ yields the lowest rollout error, significantly outperforming both higher noise levels and the noise-free setting. This suggests that controlled noise injection helps regularize the dynamics model and improves long-term stability.

This highlights a trade-off between short-term accuracy and long-term rollout stability.

Based on these results, we adopt a noise amplitude of $2 \times 10^{-3}$ for DeformingBeam experiments when evaluating

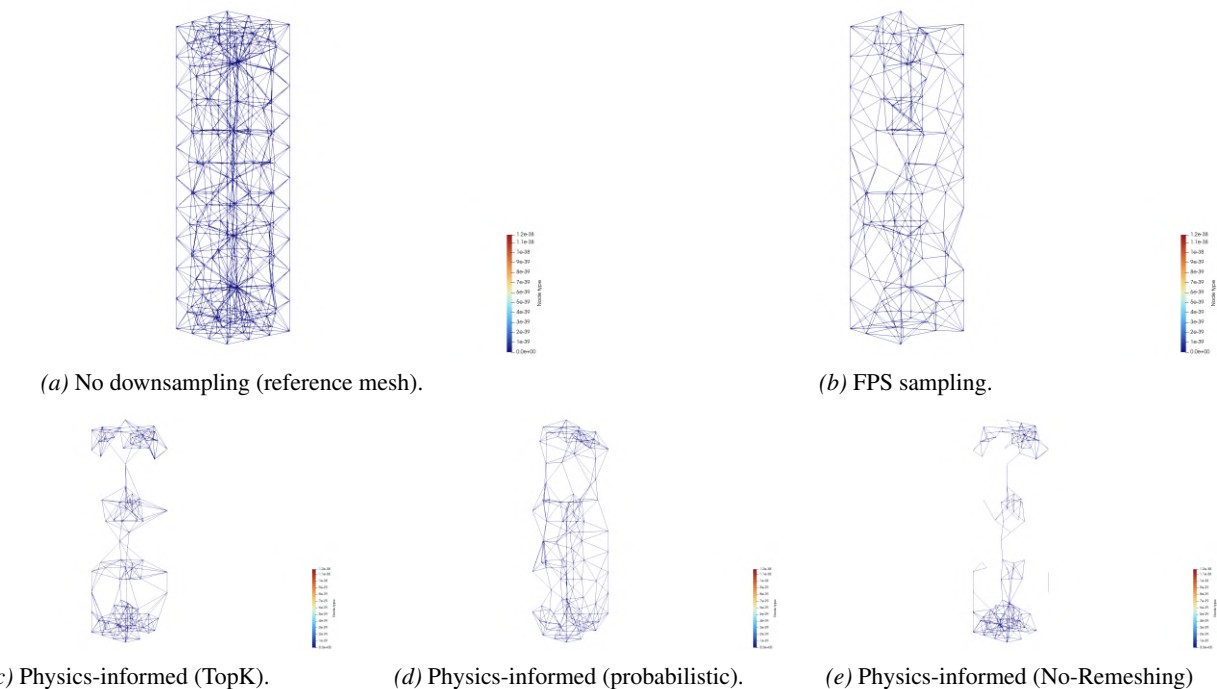

*(a)* No downsampling (reference mesh).          *(b)* FPS sampling.

*(c)* Physics-informed (TopK).          *(d)* Physics-informed (probabilistic).          *(e)* Physics-informed (No-Remeshing)

*Figure 5.* Qualitative comparison of node selection for *BeamSimple* under different sampling strategies. Physics-informed sampling retains dynamically active regions while preserving global connectivity. (step=2)

long-horizon rollouts, as this regime best reflects the intended use of learned simulators.

*Table 8.* Trade-off between rollout stability and single-step accuracy for different noise amplitudes on *DeformingBeam*.

| Noise amplitude | Rollout RMSE ↓ | 1-step RMSE ↓ |
|---|---|---|
| 0.02 | 0.1994 | 0.0007 |
| **0.002** | **0.0398** | 0.000427 |
| 0.0002 | 0.2857 | 0.00033 |
| 0.0 | 0.1903 | **0.000278** |

### A.3. World Edges for Contact Modeling

Datasets such as DEFORMINGPLATE or SPINDLEUPSETTING involve contact between the deformable workpiece and external rigid tools (e.g., dies or punches). To represent these interactions within a graph based surrogate model, we use *world edges* from (Pfaff et al., 2021), which encode interactions between mesh nodes and external bodies. At each time step, nodes within a fixed distance of an external object are connected to a virtual world node representing the tool, allowing contact effects to be injected during message passing without explicitly meshing the tools. The distance threshold is set to $3 \times 10^{-3}$ for DEFORMINGPLATE and 3 for SPINDLEUPSETTING, reflecting their respective spatial scales. World edges are processed jointly with mesh edges, while world nodes remain fixed and are not updated.

## B. Physics Residuals for Node Scoring

Our physics-informed sampling relies on a nodal score $s_i$ derived from the local violation of the governing solid mechanics equations. This appendix summarizes the constitutive laws (Nemer et al., 2021) and residual forms used in our datasets.

## B.1. Balance of linear momentum

Let $\Omega_t \subset \mathbb{R}^3$ denote the current configuration at time $t$, with outward normal $\mathbf{n}$ on $\partial\Omega_t$. Denote the displacement $\boldsymbol{u}$, velocity $\boldsymbol{v} = \dot{\boldsymbol{u}}$, density $\rho$, body force $\boldsymbol{b}$, and Cauchy stress $\boldsymbol{\sigma}$.

Depending on the physical regime, we use different residual forms: for quasi-static hyperelasticity (DeformingPlate) we use the Lagrangian residual $\boldsymbol{r} = \mathrm{Div}_X \, \mathbf{P}$. while for transient and elasto-plastic regimes we use the momentum-balance residual in (29).

The local form of the balance of linear momentum is:

$$\rho\ddot{\boldsymbol{u}} = \nabla \cdot \boldsymbol{\sigma} + \rho\boldsymbol{b}. \tag{17}$$

**Quasi-static setting.** When inertia is neglected:

$$\nabla \cdot \boldsymbol{\sigma} + \rho\mathbf{b} = \mathbf{0}. \tag{18}$$

## B.2. Finite-strain kinematics and stress mapping

Let $\mathbf{X}$ be the reference position and $\mathbf{x} = \boldsymbol{\varphi}(\mathbf{X}, t)$ the current position. The deformation gradient is:

$$\mathbf{F} = \frac{\partial\mathbf{x}}{\partial\mathbf{X}}, \qquad J = \det(\mathbf{F}), \qquad \mathbf{C} = \mathbf{F}^\top\mathbf{F}, \qquad \mathbf{B} = \mathbf{F}\mathbf{F}^\top. \tag{19}$$

Stress measures are related by:

$$\boldsymbol{\sigma} = \frac{1}{J}\mathbf{P}\mathbf{F}^\top. \tag{20}$$

## B.3. Neo-Hookean hyperelasticity and elastic residual (DeformingPlate)

For the quasi-static hyperelastic setting, we define the physics residual directly in the reference (Lagrangian) configuration using the first Piola–Kirchhoff stress. The formulation depends only on the displacement field $\boldsymbol{u}$.

**Kinematics.** Let $\boldsymbol{u}(\boldsymbol{X})$ denote the displacement with respect to the reference configuration. The deformation gradient is computed as

$$\mathbf{F} = \mathbf{I} + \nabla_X\boldsymbol{u}, \qquad J = \det(\mathbf{F}), \tag{21}$$

where $\nabla_X$ denotes the gradient with respect to the reference coordinates.

We further define the right Cauchy–Green tensor

$$\mathbf{C} = \mathbf{F}^\top\mathbf{F}, \qquad I_1 = \mathrm{tr}(\mathbf{C}). \tag{22}$$

**Constitutive law.** We adopt a compressible Neo-Hookean model with deviatoric–volumetric split. The first Piola–Kirchhoff stress is given by

$$\mathbf{P} = \mu J^{-2/3}\left(\mathbf{F} - \frac{I_1}{3}\mathbf{F}^{-\top}\right) + \kappa\ln(J)\mathbf{F}^{-\top}, \tag{23}$$

where $\mu$ is the shear modulus and $\kappa$ the bulk modulus.

**Elastic residual.** In the quasi-static regime, inertia is neglected and the elastic residual is defined as the divergence of the first Piola–Kirchhoff stress:

$$\boldsymbol{r} = \nabla_X \cdot \mathbf{P}. \tag{24}$$

The physics-informed node score used for coarsening is defined as

$$s_i = \|\boldsymbol{r}_i\|_2. \tag{25}$$

## B.4. Finite-strain elasto-plasticity (SpindleUpsetting)

For metal forming, plasticity is required. We consider the standard multiplicative split:

$$\mathbf{F} = \mathbf{F}_e\mathbf{F}_p, \tag{26}$$

with elastic part $\mathbf{F}_e$ and plastic part $\mathbf{F}_p$.

Elastic stress is obtained from a hyperelastic potential $\Psi(\mathbf{C}_e)$ with $\mathbf{C}_e = \mathbf{F}_e^\top\mathbf{F}_e$:

$$\mathbf{S}_e = 2\frac{\partial\Psi}{\partial\mathbf{C}_e}, \qquad \boldsymbol{\sigma} = \frac{1}{J}\mathbf{F}_e\mathbf{S}_e\mathbf{F}_e^\top. \tag{27}$$

**Yield criterion (J2 plasticity).** A common von Mises yield function is:

$$\Phi(\boldsymbol{\sigma}, \kappa) = \|\mathrm{dev}(\boldsymbol{\sigma})\| - \sqrt{\frac{2}{3}}\sigma_y(\kappa) \leq 0, \tag{28}$$

where $\kappa$ is a hardening variable. The plastic evolution follows an associated flow rule and is handled by the simulator through return-mapping.

## B.5. Residual based nodal score used for sampling

To construct physics-informed coarse graphs, we define a nodal residual vector $\mathbf{r}_i$ derived from the governing balance laws of solid mechanics. At time step $t$, the residual at node $i$ is evaluated on the *predicted physical state* and given by

$$\mathbf{r}_i = \rho_i\ddot{\mathbf{u}}_i - (\nabla\cdot\boldsymbol{\sigma})_i - \rho_i\mathbf{b}_i, \tag{29}$$

where $\mathbf{u}_i$ denotes the displacement, $\boldsymbol{\sigma}$ the Cauchy stress tensor, $\rho_i$ the material density, and $\mathbf{b}_i$ the body force.

The scalar physics score used for node ranking or sampling is defined as the $\ell_2$ norm of the residual:

$$s_i = \|\mathbf{r}_i\|_2. \tag{30}$$

High values of $s_i$ indicate regions of strong physical imbalance, such as stress concentrations, large deformation zones, or contact interfaces, which are preferentially retained during multigrid coarsening.

## B.6. Implementation details for physics residual computation

Our physics-informed coarsening strategy relies on nodal residual quantities derived from the governing balance laws (see Appendix B), including the divergence of the Cauchy stress $(\nabla\cdot\boldsymbol{\sigma})_i$, inertial accelerations $\ddot{\mathbf{u}}_i$ (for transient regimes), and body forces $\rho_i\mathbf{b}_i$.

**Importantly, all residual quantities are computed directly from the model's decoded predictions at each timestep.** Spatial derivatives required for the residual are evaluated using fixed discretization operators defined on the mesh, ensuring that no ground-truth, solver output, or future information is accessed during inference. In particular, we do not approximate spatial derivatives using graph based finite differences unless explicitly stated.

**Graph based divergence operator** We implemented a graph based finite-difference divergence operator $\mathrm{Div}_{\mathcal{G}}(\cdot)$, which estimates spatial derivatives directly from the mesh graph. Given node positions $\{\mathbf{x}_i\}$ and edges $(i, j) \in \mathcal{E}$, edgewise gradient contributions are computed by projecting field differences onto relative displacements and weighting by inverse squared distance, then aggregated at nodes:

$$\nabla_{\mathcal{G}}\mathbf{u}_i \approx \frac{\sum_{j\in\mathcal{N}(i)} w_{ij}\,(\mathbf{u}_j - \mathbf{u}_i)\,(\mathbf{x}_j - \mathbf{x}_i)^\top/\|\mathbf{x}_j - \mathbf{x}_i\|^2}{\sum_{j\in\mathcal{N}(i)} w_{ij}}, \qquad w_{ij} = \frac{1}{\|\mathbf{x}_j - \mathbf{x}_i\|^2 + \varepsilon}.$$

The divergence is obtained as the trace of the estimated Jacobian, $\mathrm{Div}_{\mathcal{G}}(\mathbf{u})_i = \mathrm{tr}(\nabla_{\mathcal{G}}\mathbf{u}_i)$.

**Acceleration term.** For transient datasets (e.g., BEAMSIMPLE), the residual includes inertia and requires the nodal acceleration $\ddot{\mathbf{u}}_i$. Acceleration is estimated from the *predicted displacement fields* using a backward finite-difference time discretization. Let $\Delta t$ denote the timestep size. For each node $i$, we approximate

$$\dot{\mathbf{u}}_i^{\,t} \approx \frac{\hat{\mathbf{u}}_i^{\,t} - \hat{\mathbf{u}}_i^{\,t-1}}{\Delta t}, \qquad \ddot{\mathbf{u}}_i^{\,t} \approx \frac{\hat{\mathbf{u}}_i^{\,t} - 2\hat{\mathbf{u}}_i^{\,t-1} + \hat{\mathbf{u}}_i^{\,t-2}}{\Delta t^2}. \tag{31}$$

For quasi-static datasets (e.g., DEFORMINGPLATE), inertia is neglected by construction, and the residual score omits the acceleration term ($\ddot{\mathbf{u}}_i = \mathbf{0}$).

