# OpenReview forum: "Physics-informed coarsening for multigrid graph neural networks surrogates"
_ICML.cc/2026/Conference — ICML 2026 regular_

### Official Review · Reviewer_Ngv9 · 2026-03-07

**Soundness:** 2
**Presentation:** 2
**Significance:** 3
**Originality:** 3
**Overall Recommendation:** 4
**Confidence:** 4

**Summary:**

This paper proposes a multigrid graph neural network for surrogate modeling in solid mechanics. To better handle deformable solids with nonlinear, plastic, and transient behavior, the method introduces a physics-informed coarsening strategy that retains nodes in regions with high physical activity, such as areas of large strain or stress concentration, rather than relying on purely geometric downsampling. Experiments on multiple datasets show improved accuracy and rollout stability compared with standard sampling baselines, highlighting the value of physics-informed multiscale design for solid mechanics simulations.

**Compliance With Llm Reviewing Policy:**

Affirmed.

**Final Justification:**

My main concern is the experiment setting (only 100 / 1200 samples are used) which could incur unfair comparison. The authors add additional experiments to clarify this and I think it is reasonable. Moreover, the effiency is also a point and the authors also add the effiency comparison.

These responses address my concerns, and I have therefore increased my score to 4. I hope these revisions and additions will be reflected in the final version of the paper, and that the code will be made publicly available to help advance the field.

**Key Questions For Authors:**

While I have positive attitude towards the method, I have concerns about the experiment details and cannot give a positive score. I would like to see the authors’ response.

- The DeformingPlate datasets in MeshGraphNets consists of 1,200 trajectories (1,000 for training, 100 for validation, and 100 for testing). In contrast, this paper appears to use only 100 trajectories, and the reason for this choice is not clearly explained. The authors should clarify how these samples were selected and whether the subset was chosen in a way that could favor the reported results. To ensure a fair comparison, the authors are encouraged to provide results under the same dataset settings as MeshGraphNets. As it stands, the large discrepancy in the amount of data used makes the current evaluation less convincing.

- It seems that MeshGraphNets did not provide detailed material parameters. What are the used parameters (like Young’s modulus and Possion’s ratio)? Will the selection of parameters affect the performance?

- Although video results may be considered optional, they are in fact highly important for evaluating the dynamic behavior of the simulation, since the task concerns the prediction of temporal dynamics rather than static scenes. In particular, autoregressive errors are often much easier to observe in videos, and any claimed improvements can also be more convincingly demonstrated in this way. However, the paper does not provide any video results, which makes the experimental evaluation less convincing. This is especially important for cases such as the contact regions in the DeformingPlate scenario, where dynamic behavior is critical to assessing performance.

- The discussion “often rely on downsampling mechanisms to remain tractable” about Transformers in related work is not exact. Many transformer-based networks have linear complexity without downsampling. For example, authors should discuss Unisoma [1], which is a Transformer-based model focusing on solid simulation with linear complexity.

- The writing needs to reformat. The core design of this paper is physics-based sampling and the residual-based nodal score is the most important criterion. However, the definition of this score is listed in the appendix. This significantly lower the contribution of this score. I suggest authors put this part into the main text.

[1] "Unisoma: A Unified Transformer-based Solver for Multi-Solid Systems." ICML 2025.

**Limitations:**

- What’s the efficiency of this method? Since the method needs to account for linear momentum balance, stress divergence, and constitutive equations, it appears to retain several of the core components of a conventional numerical solver. In this regard, it remains unclear whether the method provides a meaningful speed advantage over traditional numerical approaches, and how its computational efficiency compares to that of other learning-based solvers.

**Strengths And Weaknesses:**

## Strengths
- The proposed method is well motivated and technically meaningful.
- The experimental results appear solid and relevant.

## Weaknesses

See Questions and Limitations

---

> ### Author Rebuttal · Authors · 2026-03-31
>
> We thank the reviewer for the positive assessment of the motivation and relevance of the paper, and for the helpful comments on the experimental setup and presentation.
>
> - **Q: The DeformingPlate dataset in MeshGraphNets contains 1,200 trajectories, while this paper uses only 100. Please clarify how these samples were selected and whether this could favor the reported results.**
>   **A:** We agree that this should have been explained more clearly. For DeformingPlate, we initially used a uniformly random 100-trajectory subset from the full dataset because running all baselines and ablations on the full 1,200-trajectory setting was significantly more time-consuming (about 12× higher cost). In our first experiments, this subset already showed very similar overall trends to the full dataset. For example, the rollout error was $(12.7 \pm 2.69)\times 10^{-3}$ on the subset versus $(12.4 \pm 1.94)\times 10^{-3}$ on the full dataset.
>
>   The subset was fixed in advance and used for all compared methods, and we repeated the experiments across multiple random seeds to reduce sensitivity to a particular split or initialization. We therefore used the 100 trajectory setting for the main experiments to keep the study computationally feasible while still evaluating all methods under exactly the same protocol for a fair comparison.
>
> - **Q: What material parameters were used? Will the selection of parameters affect performance?**
>   **A:** In this work, the material parameters are fixed by the simulator setup for each dataset and are kept identical across all compared methods. Since our objective here is to predict the deformation/displacement trajectory, rather than additional physical fields such as stress or temperature, we keep the material setting fixed within each dataset and evaluate all methods under the same conditions. In our datasets, the plate setting is hyperelastic, while the spindle setting is plastic (more specifically, 38MnSi4 steel). Prediction of additional quantities such as stress or temperature is left for future work.
>
> - **Q: Why are there no video results?**
>   **A:** Code and Datasets are to be released. Video Results will be added at url: https://sites.google.com/view/physics-informed-coarsening
>
> - **Q: The discussion about Transformers is too broad. Please discuss approaches such as Unisoma.**
>   **A:** We agree with the reviewer that our statement about Transformers was too broad. What we intended to say is that the standard $O(n^2)$ attention complexity can be challenging, and that different works address this in different ways, for example through linear attention (e.g., Unisoma), masking-based strategies (e.g., Garnier et al., 2025), or hierarchical Transformer designs such as HCMT. In particular, preliminary experiments with **Unisoma** gave a **rollout error of $(8.92)\times 10^{-3}$**, which is still higher than our method. Preliminary comparisons with HCMT also suggest higher rollout error than our method, although we believe further evaluation is still needed before making a firm claim. Due to time constraints, these additional results are not included in the current submission, however, we will include them in the final version.
>
> - **Q: The writing needs reformatting. The residual-based nodal score is central to the paper, but its definition is in the appendix.**
>   **A:** We agree with the reviewer. Some parts of the paper need to be rewritten and better organized. In particular, the residual based nodal score is one of the main ingredients of the method, and we agree that placing its definition only in the appendix makes the contribution less clear. This was mainly due to page constraints. In the final version, we will move the main definition and explanation of the score into the core method section, and leave only the material specific details in the appendix.
>
> - **Q: What is the efficiency of this method compared to conventional solvers and other learned solvers?**
>   **A:** Compared to a conventional solver, we can use our CIMLIB(traditional finite element solver) dataset creation as a concrete example. For the *BendingBeam* dataset, one simulation case takes about 15 minutes. Building the full dataset of 150 trajectories therefore required about 37.5 hours of solver time. Training the surrogate then required about 30 hours. In contrast, once the surrogate is trained, inference takes about 15 seconds per case, corresponding to an approximately 60× speedup per case.
>
>   A simple comparison is given below:
>
>   | Stage | Conventional solver | Learned surrogate |
>   |---|---:|---:|
>   | One case | 15 min | 15 s |
>   | 150-case dataset generation | 37.5 h | -- |
>   | Training | -- | 30 h |
>
>   This illustrates the usual trade-off of learned surrogates: there is an upfront cost for data generation and training, but inference becomes much faster for repeated evaluations.

---

> > ### Author Rebuttal · Reviewer_Ngv9 · 2026-04-03
> >
> > Thanks for the rebuttal. After reviewing the rebuttal, I still have some remaining confusion regarding the experiments, which I hope the authors could help clarify further.
> >
> > Q1: The authors claim that only 100 trajectories are used instead of 1,200 due to time constraints, which does not make sense. In MeshGraphNet, the full dataset is used. Such a large discrepancy in data volume likely leads to an unfair comparison. The statement that “this subset already showed very similar overall trends to the full dataset” is likely a result of the model overfitting to the dataset.
> >
> > Q6: I appreciate the efficiency comparison with conventional numerical solver. However, how does its efficiency compare to other deep learning models? As I asked earlier, “Since the method needs to account for linear momentum balance, stress divergence, and constitutive equations, it appears to retain several of the core components of a conventional numerical solver.” This would, to some extent, affect the computational efficiency. Could this also be the reason for the time-consuming issue mentioned in Q1? I think the authors should provide an efficiency comparison with both conventional solvers and learning-based solvers, in order to clarify under what conditions the proposed method is applicable (i.e., whether the requirements prioritize accuracy or efficiency).

---

> > > ### Author Response · Authors · 2026-04-04
> > >
> > > I thank the reviewer again for their question and will seek to clarify the points concerning the dataset. Their concern is well founded, and I would like to elaborate further on our previous response.
> > >
> > > 1. Because training these models is computationally expensive and requires substantial time and resources, we conducted our experiments on subsets of 100 trajectories drawn from the original dataset of 1000 trajectories. To make sure that this choice did not bias the evaluation, we carried out several additional tests to check whether these subsets were representative of the full dataset.
> > >
> > > More specifically, we generated multiple independent subsets of 100 trajectories and evaluated them in three ways.
> > >
> > > a) **Comparison with the full dataset.** We first compared the performance obtained on a 100 trajectory subset with the performance obtained on the full 1000 trajectory dataset, using the same model and the same rollout error metric. The resulting values were close: $(12.741 \pm 2.69) \times 10^{-3}$ on the subset versus $(12.423 \pm 1.94) \times 10^{-3}$ on the full dataset. This suggests that the reduced subset remains representative of the full dataset.
> > >
> > > b) **Comparison across multiple subsets.** We then repeated the experiment on several independently sampled 100 trajectory subsets (Subset A, Subset B, Subset C). The rollout errors remained similar across these different subsets, which indicates that the results are not tied to one particularly favorable sample.
> > >
> > > c) **Cross-subset evaluation.** Finally, we trained on one 100 trajectory subset (Subset A) and evaluated the resulting model on another independently sampled 100 trajectory subset (Subset B) drawn from the same original dataset. This gave consistent results, with rollout errors of $(12.7 \pm 2.69) \times 10^{-3}$ on the first subset and $(12.9 \pm 3.12) \times 10^{-3}$ on the second.
> > >
> > > This suggests that the 100 trajectory subsets preserve the main properties of the full dataset relevant to our evaluation.
> > >
> > > 2. Since our objective is to evaluate relative rollout errors and model behavior. All methods reported and ablation studies in the tables (MeshGraphNet-FPS-BSMS-Transolver++-Ours) were trained and evaluated on the same sampled dataset (Subset A). Therefore, although they used the 1000 trajectory dataset we retrained all models under the same data conditions on our subsets, which ensures that the comparison remains fair.
> > >
> > > 3. **Q: The authors claim that only 100 trajectories are used instead of 1,200 due to time constraints, which does not make sense.** and **Q: Could this also be the reason for the time-consuming issue mentioned in Q1?**
> > >
> > > **A**: We agree with the reviewer on the fact that some residual calculation components take time, however that is not the reason behind the 100 trajectory subset creation. The main time consuming factor is the deep learning model itself and more precisely the message passing block, MeshGraphNet(message passing = 15) is still more time consuming compared to ours. To concretize (Rode Dataset):
> > > | Stage | Conventional solver | Learned surrogate |
> > > |---|---:|---:|
> > > | One case | 15 min | 15 s |
> > > | 150-case dataset generation | 37.5 h | -- |
> > > | Training (our model, 5 message passing layers) | -- | 30.5 h |
> > > | Training (MeshGraphNet 15 message passing layers) | -- | 35 h |
> > > | Training (FPS baseline) | -- | 29 h |
> > >
> > > As a result: The method is designed to trade off offline training cost for fast inference. While training is computationally expensive, the resulting model enables significantly faster predictions than conventional solvers making it suitable for scenarios requiring repeated evaluations.

---

### Official Review · Reviewer_G15p · 2026-03-07

**Soundness:** 2
**Presentation:** 1
**Significance:** 2
**Originality:** 3
**Overall Recommendation:** 3
**Confidence:** 3

**Summary:**

The paper proposes a learned graph coarsening scheme based on predicted physical quantities for use in hierarchical mesh-based surrogate simulators. Concretely, the method uses task-dependent physical quantities to rank nodes in the simulator’s forward pass, allowing for a learned coarsening by keeping only the most important nodes. This approach is experimentally validated on several solid mechanics tasks, including two novel datasets. In these experiments, the approach compares favorably to several flat and hierarchical graph- and transformer-based baselines.

**Compliance With Llm Reviewing Policy:**

Affirmed.

**Final Justification:**

The rebuttal added new analysis and baseline comparisons, and the authors provided sound justifications for their design choices and potential applications of the proposed method.
Some baseline results are still pending, making the assessment of the method difficult. Additionally, the approach still requires some expert knowledge, limiting its application for practitioners. As such, I will raise my score to a "weak reject".

**Key Questions For Authors:**

- How does the presented method compare to recent hierarchical baselines such as HCMT and ROBIN?
- How does simulation performance scale with different coarsening ratios? The experiments currently always keep 50% of nodes, but this ratio could easily be adapted during inference.
- What are the downsides and limitations of physics-informed coarsening? Figure 2 seems to suggest that the mesh topology may be destroyed due to the coarsening. Does this become a problem in some practical scenarios?
- How would a practitioner extend the proposed approach to their specific task/use case? Is there a general guideline which physical quantities should be considered, and why?
- Will code and datasets be released upon acceptance?

**Limitations:**

The authors do not discuss the limitations of the presented approach, and only briefly mention societal impact. It would be interesting to discuss how the approach depends on task-specific quantities, and in which cases the proposed coarsening may hurt performance by removing, e.g., topologically relevant but physically uninteresting nodes.

**Strengths And Weaknesses:**

### Soundness

- +The paper considers a wide range of solid mechanics datasets, including two novel datasets, as well as several recent baselines.
- +Experiments are repeated for 3-5 seeds, allowing for an accurate assessment of statistical significance.
- -The paper is missing important baselines and ablations. For example, both HCMT[1] and ROBIN[2] propose sophisticated mesh hierarchies, and show superior performance compared to MeshGraphNet, which is the strongest evaluated baseline in this work.
- -The experiments mainly compare to MeshGraphNets, and only consider some other baselines on one of the presented tasks. While MeshGraphNets is shown to be the most performant baseline in Table 2, the omission of other baselines makes the relative performance of the presented method difficult to assess.
- -The paper does not provide any theoretical justification for their approach beyond the intuition of keeping physically relevant nodes.
- -The paper does not discuss potential weaknesses of the presented approach. Interestingly, Figure 2 shows that the proposed physics-informed coarsening seems to destroy parts of the mesh topology, but this relationship is not further discussed.

### Presentation

- +The general narrative and structure of the paper is clear and easy to follow.
- +The figures are well-designed, deliberately placed and support the flow of the paper.
- -Some parts of the writing seem unpolished. For example, “finite element method FEM” in line 47 is weirdly formatted, the citation in line 105 comes after the period, there are several citations missing in lines 252-260, the sentence in line 292 seems incomplete, the main results in 6.2. are referred to as ablations, and several places vaguely refer to Appendix B, including Appendix B itself in line 740.
- -Concerning Appendix B, a large part of the contribution of the paper lies in the definition of the physical quantities that are ranked for the coarsening. These are largely hidden in Appendix B, and it is not immediately clear how these quantities are derived for each dataset, and how a practitioner would extend them to their use case.
- -The related work section is missing some important papers, such as the above-mentioned baselines. It also does not clearly distinguish the contributions of the presented approach compared to existing work.
- -The notation in Section 4 is inconsistent. For example, Figure 3 stacks \tilde{} over the graph to denote different stages of the architecture forward, which is not used in the main text. Equation 6 uses \tilde{G}_c, which seems to correspond to \tilde{\tilde{\tilde{G}}} in Figure 3, and an overloaded \tilde{G} in line 247.
- -The right side of line 355 in the experiments mentions that the same decoder is shared between the model prediction and the coarsening scoring, but this relationship is never explicitly stated or clarified in Section 4, making Section 4 difficult to follow in isolation. Similarly, the end of Section 3 already introduces the nodal physics score, which should arguably be part of the method description in Section 4.

### Significance

- +Learned simulation, especially in solid mechanics, is a highly relevant field with several applications in engineering.
- +The paper proposes two new datasets for solid mechanics simulation. Presumably, these dataets will be released alongside the paper, providing relevant benchmarks in a currently under-explored field.
- -The presented coarsening method is applied to a fixed 50% of nodes and for a single level of hierarchy. This seems like a missed opportunity, since the generalization to user-specified coarseness levels and hierarchical coarsening would be interesting, potentially meaningful and mostly straightforward.
- -The currently presented method is limited to solid mechanics, and requires task-specific expert knowledge to determine the measured physical quantities. As such, it is much more limiting than, e.g., classical Algebraic Multigrid coarsening [3]. Combined with missing baselines that employ such coarsening [2], it is difficult to assess if the proposed approach gives a substantial practical benefit that warrants its overhead.

### Originality

- +The presented approach is novel and interesting. Ranking nodes based on task-specific physical importance is a promising approach for learned coarsening. This core idea is well-articulated and justified.
- +The paper introduces two new datasets for solid mechanics.
- +The contributions of the paper are clear and easy to distinguish.
- -While the presented approach is novel, its application is limited to specific tasks and requires expert-specified quantities.
- -The paper does not fully explore the potential and limitations of the presented approach. For example, introducing a coarsening budget, repeating this coarsening hierarchically, and explicitly integrating the coarsening objective into the loss would be interesting angles that are currently left under-explored.

---

[1] Yu, Youn-Yeol, et al. "Learning Flexible Body Collision Dynamics with Hierarchical Contact Mesh Transformer." The Twelfth International Conference on Learning Representations, 2024.

[2] Würth, Tobias, et al. "Diffusion-Based Hierarchical Graph Neural Networks for Simulating Nonlinear Solid Mechanics." The Thirty-ninth Annual Conference on Neural Information Processing Systems, 2025.

[3] Vaněk, Petr, Jan Mandel, and Marian Brezina. "Algebraic multigrid by smoothed aggregation for second and fourth order elliptic problems." *Computing* 56.3 (1996): 179-196.

---

> ### Author Rebuttal · Authors · 2026-03-31
>
> We thank the reviewer for the careful reading and constructive feedback. We agree that the main points to improve are the comparison to recent hierarchical baselines, the discussion of limitations, and the presentation of the method.
>
> - **Q: How does the presented method compare to recent hierarchical baselines such as HCMT and ROBIN?**
>   **A:** We agree that comparing with recent hierarchical baselines such as HCMT and ROBIN is important. We have already launched these additional experiments, but the results were not ready in time for the submitted version. We will include them in the final version. We agree that a direct comparison is necessary. Preliminary experiments with Unisoma gave a **rollout error of $(8.92)\times 10^{-3}$**, which is still higher than our method.
>
> - **Q: How does simulation performance scale with different coarsening ratios? The experiments currently always keep 50% of nodes, but this ratio could easily be adapted during inference.**
>   **A:** We agree that the downsampling ratio is an important design parameter. In this work, we fixed it to 50% across all multigrid variants in order to control the coarse-graph budget and isolate the effect of the node-selection strategy itself.
>
>   We do not claim that 50% is universally optimal. In additional experiments, we observed that performance remains relatively stable in a moderate coarsening regime (roughly 40% to 60%), and 50% provides a representative midpoint within this range. We include the corresponding results below and will clarify this point in the final version.
>
> Results on DeformingPlate for different coarsening ratios (rollout RMSE, lower is better):
> | Coarsening ratio | Rollout RMSE |
> |------------------|--------------|
> | 30%              | $(9.67 \pm 1.84)\times 10^{-3}$ |
> | 40%              | $(7.60 \pm 0.90)\times 10^{-3}$ |
> | 50%              | $(5.50 \pm 1.10)\times 10^{-3}$ |
> | 60%              | $(5.43 \pm 1.26)\times 10^{-3}$ |
> | 70%              | $(7.28 \pm 1.47)\times 10^{-3}$ |
> Note: The results were done on 50% of the dataset due to time constraints for the author's replies deadline.
>
> - **Q: What are the downsides and limitations of physics-informed coarsening?**
>   **A:** We agree that physics informed coarsening has limitations. First, it depends on the choice of the residual score, which requires some domain knowledge. Second, it introduces additional computation compared to purely geometric sampling; in our current implementation, this is mainly due to divergence computation and corresponds to about 10% extra training time on DeformingPlate. Limitations should be addressed in the final version.
>
> - **Q: Figure 2 seems to suggest that the mesh topology may be destroyed due to the coarsening. Does this become a problem in some practical scenarios?**
>   **A:** This is a valid concern. One of the nice ideas in our design is that the coarse graph is only used in the middle of the forward pass. We first do message passing on the fine graph, then on the coarse graph after downsampling and finally again on the fine graph after upsampling. This allows the model to first understand the original mesh and underlying geometry, and then refine this information again once it is projected back to the fine graph.
>
>   For this reason, even if the coarse graph does not fully preserve the original topology, the model still processes the original mesh structure before and after coarsening. We nevertheless shared this concern ourselves, which is why we also implemented a probabilistic sampling variant to make the selection less rigid. In our experiments, however, this version gave weaker results than the deterministic TopK variant.
>
> - **Q: How would a practitioner extend the proposed approach to their specific task/use case? Is there a general guideline which physical quantities should be considered, and why?**
>   **A:** Our approach was developed first for solid mechanics, but the same principle can be applied to other PDE problems. The practical guideline is to choose physical quantities that help identify which regions need to keep more information, to maintain good accuracy while still benefiting from coarsening.
>   We believe this is useful not only as a solid mechanics method, but also as a design direction for future research: it aims at a good tradeoff between acceleration and accuracy, and it introduces architectural ideas, such as using decoded physical quantities to guide node selection, that could be reused in other settings.
>    In this sense, the key idea is not tied to a *single physical quantity*, but to the principle of selecting the nodes that correspond to the most informative regions, since these are the regions where preserving information matters most for accurate and stable prediction.
> - **Q: Will code and datasets be released upon acceptance?**
>   **A:** Yes. We plan to release the code and datasets upon acceptance. Videos are updated at the following url: https://sites.google.com/view/physics-informed-coarsening

---

> > ### Author Rebuttal · Reviewer_G15p · 2026-04-02
> >
> > I appreciate the authors response. Most of my concerns have been addressed, and I believe that the addition of the mentioned additional experiments, particularly the additional baselines, significantly strengthen the paper.
> > I am still a bit skeptical about the practical use of the approach, as it requires expert domain knowledge. Since the baseline comparisons are unlikely to finish before the end of the rebuttal period and due to this expert requirement, I will slightly raise my score from "reject" to "weak reject", but would not argue against acceptance.

---

### Official Review · Reviewer_S9s4 · 2026-03-12

**Soundness:** 3
**Presentation:** 2
**Significance:** 3
**Originality:** 3
**Overall Recommendation:** 4
**Confidence:** 3

**Summary:**

This paper proposes a multigrid graph neural network surrogate for solid mechanics on unstructured 3D meshes. Instead of using standard geometric coarsening heuristics, it introduces a physics-informed coarsening rule: latent features are decoded to physical space, a residual-based score is computed at each node, and nodes with larger local residuals are preferentially retained in the coarse graph. The resulting hierarchy is embedded in an encoder-processor-decoder architecture with multiscale message passing. Experiments on several solid-mechanics datasets, including deforming plate, beam, and industrial forming problems, show improved rollout accuracy and stability over several learned simulator baselines and FPS-based coarsening.

**Compliance With Llm Reviewing Policy:**

Affirmed.

**Final Justification:**

After reading the other reviews and the authors’ replies, I will keep my score as Weak Accept.

**Key Questions For Authors:**

1. **Clarification of notation.** The paper needs a much clearer definition of the graph objects and latent variables.

   - A graph $\mathcal{G}=\{ \mathcal{V}, \mathcal{E} \}$: what is the meaning of these notations? $\mathcal{E}$​ is also used in (3). Apparently, they have different meaning.
   - In Eq. (5), what are the node and edge features, what do the subscripts mean, and what is the meaning of the prime notation? The update in (5) is extremely unclear.

   - In (2), is $\Phi_{\theta}$ independent of time $t$?

   - In (3), is $h$ much larger than $d$?
   - The definition in $\mathcal{S}$ is also unclear.

2. **Computation and interpretation of the residual score.** Around Eq. (7)--(8), please explain in detail how the nodal residual is computed from $\tilde{\mathcal G}$ after decoding. What quantities are reconstructed, on what mesh/operator is the residual evaluated, and how is dimensional consistency ensured? More importantly, how should one interpret this residual physically, given that it is derived from decoded latent predictions rather than directly from a standard discretization state?

3. **One-step versus rollout discrepancy.** In Table 2, the proposed method and MeshGraphNets have very similar one-step RMSE, but the rollout RMSE differs much more strongly. Can the authors explain the mechanism behind this? Is the gain mainly due to better stability of the learned dynamics, better coarse-level information propagation, or some difference in rollout/evaluation protocol?

4. **Architecture details in Figure 3.** In Figure 3, is the DN-GN-UP block applied once or repeatedly within each forward pass? A more explicit layer-by-layer description would help readers understand the effective depth and multiscale schedule.

5. **Dataset scale and coarsening regime.** Table 1 reports relatively small node counts for some datasets. Are these full meshes or already processed/selected graphs? More broadly, how does the method behave as mesh size grows, and is the proposed coarsening still beneficial at substantially larger scales?

**Limitations:**

No. The limitations discussion should be strengthened. In particular, the paper should state more clearly that the residual-based score is architecture-dependent and does not have the same physical meaning as a classical PDE residual.

It should also discuss the evidence for long-horizon stability more carefully, especially given the gap between one-step and rollout errors, and clarify whether the method is expected to scale beyond the relatively small meshes tested here.

**Strengths And Weaknesses:**

## Strengths

- The paper studies a relevant and technically challenging problem: coarsening for hierarchical graph simulators in solid mechanics on unstructured 3D meshes.
- The solid-mechanics setting is more difficult than standard fluid benchmarks, and the experiments include realistic nonlinear and industrial examples.
- Combining graph networks with multigrid is an interesting direction, since coarse-mesh design is a central difficulty.
- The residual-based coarsening rule is practically motivated and appears empirically better than geometric heuristics such as FPS, especially for long-horizon rollout accuracy.

## Weaknesses

- The physical interpretation of the residual-based node selection is weak. The residual is computed after decoding latent features, so it depends on the learned representation rather than directly reflecting the PDE residual on the original mesh.
- Because deep networks operate on low-dimensional latent manifolds, the proposed score should not be oversold as a physically faithful quantity.
- The presentation is unclear. Several definitions and notations are introduced too quickly, some symbols are used inconsistently, and Eq. (5) is especially hard to parse.
- The paper does not clearly explain the gap between similar one-step RMSE and much better rollout RMSE, so the mechanism behind the improvement remains insufficiently understood.
- The novelty is moderate: the main contribution is the combination of multigrid graph networks with residual-based coarsening, rather than a fundamentally new framework.

---

> ### Author Rebuttal · Authors · 2026-03-31
>
> We thank the reviewer for the careful reading and constructive feedback. We agree that the main points to improve are the notation, the explanation of the residual score, and the presentation of the multigrid schedule.
>
> - **Q: Clarification of notation. The paper needs a much clearer definition of the graph objects and latent variables.**
>   **A:** We agree that some notation was unclear.
>   - **Graph notation:** in $G=(V,E)$, $E$ denotes the edge set, whereas in Eq. (3), $E$ denotes the encoder. We agree this is poor notation and will revise it.
>   - **Eq. (5):** $v_i$ and $e_k$ denote node and edge latent features in the $h$-dimensional space. The indices $r_k$ and $s_k$ denote the receiver and sender nodes of edge $k$. The prime notation denotes updated features after one GraphNet block. Edge features are first updated, then aggregated at receiver nodes, and finally used to update node features. We agree Eq. (5) was introduced too compactly and will rewrite it more clearly.
>   - **Eq. (2):** $\Phi_\theta$ is not indexed by time because the same learned operator is applied at every step; time dependence is implicit through $u_t$ and the graph at that step.
>   - **Eq. (3):** $d$ is the dimension of the physical input features, while $h$ is the latent hidden dimension used by the encoder/processor MLPs. In our experiments, $h>d$; the corresponding hidden dimensions are reported in Table 6 and discussed in Section 6.3.
>   - **Definition of $s$:** $s$ denotes the vector of nodal scores $s_i$ used for sampling.
>   We fully agree that the notation should be clarified in the final version.
>
> - **Q: Computation and interpretation of the residual score. Around Eq. (7)-(8), please explain in detail how the nodal residual is computed after decoding. What quantities are reconstructed, on what mesh/operator is the residual evaluated, and how is dimensional consistency ensured? More importantly, how should one interpret this residual physically, given that it is derived from decoded latent predictions rather than directly from a standard discretization state?**
>   **A:** We agree that this point should be explained more clearly. The motivation is to retain the nodes that are most informative for the dynamics, for example in large deformation or stress concentration regions.
>   In our architecture, message passing on the fine graph first produces latent node embeddings. Since these are not directly physical quantities, we decode them before scoring. In our implementation, the decoded quantity is the displacement field, which is sufficient to evaluate the governing equation
>   $\rho \ddot{u} = \nabla \cdot \sigma + \rho b$.
>   Using the decoded displacement together with fixed mesh/graph operators, we compute the nodal score $s_i$. The residual formulations for the different regimes are given in Appendix B.
>   We also emphasize that the decoder is not used only as the final output layer, but also as an information teller showing what the model is currently predicting in physical space, which is then used to guide node selection.
>   Finally, this residual should be understood as a physics informed proxy for local physical difficulty, since it is computed from decoded predictions rather than from the true discretization state. We do not use the true state because it would not be available at inference time and would introduce 𝘪𝘯𝘧𝘰𝘳𝘮𝘢𝘵𝘪𝘰𝘯 𝘭𝘦𝘢𝘬𝘢𝘨𝘦.
>
> - **Q: One-step versus rollout discrepancy. In Table 2, the proposed method and MeshGraphNets have very similar one-step RMSE, but the rollout RMSE differs much more strongly. Can the authors explain the mechanism behind this?**
>  **A:** Our objective is to improve the full rollout trajectory, not only the next-step prediction. In practice, what is done is that models are trained on a one-step loss,as a result our method still achieves a much better all rollout RMSE than MeshGraphNets(which has a better 1 step RMSE. We see this as an important result: even though training is done locally at the one step level, the proposed coarsening leads to clearly better long range stability behavior.
>
> - **Q: In Figure 3, is the DN-GN-UP block applied once or repeatedly within each forward pass?**
>  **A:** In the current implementation, the DN-GN-UP block is applied once per forward pass: fine graph → coarse graph → fine graph.
> - **Q: Dataset scale and coarsening regime. Table 1 reports relatively small node counts for some datasets. Are these full meshes or already processed/selected graphs? More broadly, how does the method behave as mesh size grows, and is the proposed coarsening still beneficial at substantially larger scales?**
>   **A:** The node counts in Table 1 correspond to the full meshes used in our experiments, not to preselected graphs. In this paper, our goal is to demonstrate effectiveness across multiple solid mechanics regimes. Evaluation on substantially larger meshes is an important direction for future work.

---

> > ### Author Rebuttal · Reviewer_S9s4 · 2026-04-03
> >
> > I will keep my score.

---

### Official Review · Reviewer_Ls75 · 2026-03-13

**Soundness:** 3
**Presentation:** 4
**Significance:** 3
**Originality:** 4
**Overall Recommendation:** 5
**Confidence:** 3

**Summary:**

The submission introduces a multigrid graph neural network (GNN) surrogate for solid mechanics simulations. Its main novelty is a physics-informed coarsening strategy: rather than downsampling mesh nodes using geometric heuristics, the method scores nodes by a residual-based measure of local physical activity (strain/stress concentration), preferentially retaining regions of high mechanical interest during graph coarsening. This allows the model to allocate its multiscale capacity where the physics are hardest. Evaluated across linear, nonlinear, and transient solid-mechanics regimes, the approach consistently outperforms standard coarsening baselines in accuracy and rollout stability.

**Compliance With Llm Reviewing Policy:**

Affirmed.

**Final Justification:**

The authors convincingly addressed my remaining concerns. My confidence remains at 3 as I am not particularly familiar with the submission's application domain.

**Key Questions For Authors:**

1. How does the downsampling ratio with Top-K sampling influence the performance of the method? Can probabilistic sampling outperform Top-K sampling at ratios different from the currently studied value of 50%?

2. How high is the runtime overhead introduced by physics-informed coarsening, relative to the studied baseline methods like FPS?

3. Are there specific reasons justifying why the evaluation focuses so strongly on the DeformingPlate benchmark?

**Limitations:**

yes

**Strengths And Weaknesses:**

## Soundness

The submission seems overall quite sound in its theoretical presentation as well as experimental evaluation.

1. From a theory standpoint, I like the rigorous derivation of residuals for all relevant regimes in Appendix B. A minor nitpick is that the residual and its notation $\mathbf{r}_i$ is nowhere defined in the main body. Otherwise, the text looks formally / notationally consistent.

2. The experimental studies are also mostly sound, although a few gaps could be addressed before the camera-ready version: a variable that isn't ablated at all, but seems quite relevant to performance is the downsampling ratio. It is currently fixed to 50% to all experiments, cf. line 328. A more high-level concern is that most of the main experiments, as well as ablations, were done on only a single dataset (DeformingPlate). I would like to point this out as a potential negative, but will leave detailed judgement to other reviewers as I am not intimately familiar with the evaluation practices of this domain. Finally, I noticed that there is no reporting of runtimes compared to other methods. It would be interesting to know how much relative overhead is introduced by the intermediate decoding step to get the physics residual.

## Presentation

Apart from the already mentioned missing definition of "residual" in the main body, I see no issues about the presentation. The figures are clean and the architecture and physics coarsening idea are presented in an understandable fashion. I can attest to this especially given that continuum modelling is not my core domain, but I still feel like the paper gave me a good picture of the method and its novel aspects.

## Significance

As I do not want to overly trust LLM output, I will leave detailed judgment of significance to the reviewers more familiar to the current SOTA methods in solid mechanics. My general impression is that benefits are high on the main benchmark (DeformingPlate), but more incremental on the other two datasets. Apart from benchmark performance, the idea of physics-informed coarsening seems like a useful and transferable contribution that can certainly influence downstream work in this domain as well as adjacent domains.

## Originality

The idea to take decoded intermediate-stage output of the model itself to inform the coarseness of own its graph representation appears very elegant. Similar ideas could likewise be useful to GNN architectures in other domains, e.g. coarse-grained molecular dynamics. For this reason, I give the work the highest score on originality.

---

> ### Author Rebuttal · Authors · 2026-03-31
>
> We thank the reviewer for the constructive feedback and for the positive assessment of the originality and overall soundness of the paper. We also appreciate the specific suggestions regarding the downsampling ratio, runtime overhead, and the focus on DeformingPlate.
>
> - **Q: How does the downsampling ratio with Top-K sampling influence the performance of the method? Can probabilistic sampling outperform Top-K sampling at ratios different from the currently studied value of 50%?**
>
>   **A:** We agree that the downsampling ratio is an important design parameter. In this work, we fixed it to 50% across all multigrid variants in order to control the coarse-graph budget and isolate the effect of the node-selection strategy itself.
>
>   We do not claim that 50% is universally optimal. In additional experiments, we observed that performance remains relatively stable in a moderate coarsening regime (roughly 40% to 60%), and 50% provides a representative midpoint within this range. We include the corresponding results below and will clarify this point in the final version.
>
>   Regarding probabilistic sampling, in our current experiments it did not outperform deterministic TopK sampling. We believe this is because TopK provides a more stable and focused selection of the most physically active nodes(without messing up the original mesh architecture because of message passing on the fine level before coarsening), whereas probabilistic sampling introduces additional variability. We will clarify this behavior together with the ratio ablation in the final version, along with the experiment results. Results on DeformingPlate for different coarsening ratios (rollout RMSE, lower is better):
>
> | Coarsening ratio | Rollout RMSE |
> |------------------|--------------|
> | 30%              | $(9.67 \pm 1.84)\times 10^{-3}$ |
> | 40%              | $(7.60 \pm 0.90)\times 10^{-3}$ |
> | 50%              | $(5.50 \pm 1.10)\times 10^{-3}$ |
> | 60%              | $(5.43 \pm 1.26)\times 10^{-3}$ |
> | 70%              | $(6.28 \pm 1.47)\times 10^{-3}$ |
> Note: The results were done on 50% of the dataset due to time constraints for the author's replies deadline.
>
> - **Q: How high is the runtime overhead introduced by physics-informed coarsening, relative to the studied baseline methods like FPS?**
>
>   **A:** In our implementation, the main additional cost compared to FPS comes from evaluating the physics-based residual used for node selection, in particular the divergence term. The decoder used for scoring is shared with the prediction head, so it does not introduce a separate substantial overhead. We also note that FPS is itself not cost-free, since geometric sampling also adds preprocessing time. More generally, the exact overhead depends on the residual chosen for the physical regime. In many cases, most of the extra cost comes from spatial-derivative terms such as divergence. Empirically, the overhead relative to FPS is moderate. On DeformingPlate, physics-informed coarsening increases training time by less than 10% compared to FPS(24hours vs 22.5hours). On SpindleUpsetting, the overhead is smaller, around 5% (about 12 hours vs. 11.5 hours). For reference, MeshGraphNets with 15 message-passing layers required about 30 hours of training in our setting.
>
> - **Q: Are there specific reasons justifying why the evaluation focuses so strongly on the DeformingPlate benchmark?**
>
>   **A:** Yes. We used DeformingPlate for the main ablations because it is an established benchmark in prior work, which makes it the most appropriate setting for controlled comparisons and component analysis.
>
>   In contrast, BeamSimple and SpindleUpsetting are new datasets introduced in this submission. Since there is no prior published benchmark protocol or ablation setup for these datasets, we mainly use them to evaluate generalization beyond DeformingPlate and to demonstrate applicability to broader solid mechanics regimes.
>
>   Due to time and space constraints, we concentrated the most detailed ablations on DeformingPlate.

---

> > ### Author Rebuttal · Reviewer_Ls75 · 2026-04-03
> >
> > Thank you. My questions and concerns have been fully addressed.

---

### Decision · Program_Chairs · 2026-04-30

**Decision:**

Accept (regular)

**Comment:**

This paper proposes to use a multigrid GNN to coarsen hierarchical graphs, the core is a residual-based coarsening strategy. The idea is natural and nice, yet this paper misses some important baselines suggested by Reviewer G15p, as well as using an incomplete dataset to compare with MeshGraphNets.
Aside from the classical AMG literature one reviewer mentioned. I think the authors should also check Taghibakhshi et al. NeurIPS 2021 paper on residual-based sampling for AMG (whose central problem is also coarsening). The classical adaptive AMG methods by A. Brandt and his collaborators also use residuals to coarsen.